# New Insights on the Genetics of Pheochromocytoma and Paraganglioma and Its Clinical Implications

**DOI:** 10.3390/cancers14030594

**Published:** 2022-01-25

**Authors:** Sakshi Jhawar, Yasuhiro Arakawa, Suresh Kumar, Diana Varghese, Yoo Sun Kim, Nitin Roper, Fathi Elloumi, Yves Pommier, Karel Pacak, Jaydira Del Rivero

**Affiliations:** 1Life Bridge Health Center, Internal Medicine Program, Sinai Hospital of Baltimore, Baltimore, MD 21215, USA; sakshijhawar24@gmail.com; 2Developmental Therapeutics Branch, National Cancer Institute, National Institutes of Health (NIH), Bethesda, MD 20892, USA; arakawa.yasuhiro@nih.gov (Y.A.); suresh.kumar@nih.gov (S.K.); diana.varghese@nih.gov (D.V.); yoosun.kim@nih.gov (Y.S.K.); nitin.roper@nih.gov (N.R.); fathi.elloumi@nih.gov (F.E.); pommier@nih.gov (Y.P.); 3Section on Medical Neuroendocrinology, Eunice Kennedy Shriver National Institute of Child Health and Human Development, National Institutes of Health (NIH), Bethesda, MD 20892, USA; karel@mail.nih.gov

**Keywords:** pheochromocytoma, paraganglioma, genetics, germline, screening

## Abstract

**Simple Summary:**

Pheochromocytoma and paraganglioma (together PPGL) are rare neuroendocrine tumors that arise from chromaffin tissue and produce catecholamines. Approximately 40% of cases of PPGL carry a germline mutation, suggesting that they have a high degree of heritability. The underlying mutation influences the PPGL clinical presentation such as cell differentiation, specific catecholamine production, tumor location, malignant potential and genetic anticipation, which helps to better understand the clinical course and tailor treatment accordingly. Genetic testing for pheochromocytoma and paraganglioma allows an early detection of hereditary syndromes and facilitates a close follow-up of high-risk patients. In this review article, we present the most recent advances in the field of genetics and we discuss the latest guidelines on the surveillance of asymptomatic *SDHx* mutation carriers.

**Abstract:**

Pheochromocytomas (PHEOs) and paragangliomas (PGLs) are rare neuroendocrine tumors that arise from chromaffin cells. PHEOs arise from the adrenal medulla, whereas PGLs arise from the neural crest localized outside the adrenal gland. Approximately 40% of all cases of PPGLs (pheochromocytomas/paragangliomas) are associated with germline mutations and 30–40% display somatic driver mutations. The mutations associated with PPGLs can be classified into three groups. The pseudohypoxic group or cluster I includes the following genes: *SDHA*, *SDHB*, *SDHC*, *SDHD*, *SDHAF2*, *FH*, *VHL*, *IDH1/2*, *MHD2*, *EGLN1/2* and *HIF2/EPAS*; the kinase group or cluster II includes *RET*, *NF1*, *TMEM127*, *MAX* and *HRAS*; and the Wnt signaling group or cluster III includes *CSDE1* and *MAML3*. Underlying mutations can help understand the clinical presentation, overall prognosis and surveillance follow-up. Here we are discussing the new genetic insights of PPGLs.

## 1. Introduction

Pheochromocytomas (PHEOs) and paragangliomas (PGLs) are rare neuroendocrine (NE) tumors arising from chromaffin cells of the adrenal medulla and extra-adrenal ganglia, respectively. The incidence of PHEOs and PGLs (collectively PPGLs) is estimated at approximately 2–8 cases per million per year [1,2]. However, this is likely an underestimate, based upon the finding of up to 0.05–0.1% incidentally detected cases in an autopsy series [3]. PPGLs may occur at any age and they usually peak between the 3rd and 5th decade of life [4]. Patients with PPGL most commonly present with symptoms of excess catecholamine production including headache, diaphoresis, palpitations, tremors, facial pallor and hypertension. These symptoms are often paroxysmal, although persistent hypertension between these episodes is common and occurs in 50–60% patients with PPGL [5].

The field of genomics in PPGL has rapidly evolved over the past two decades. Approximately 40% of all cases of PPGLs are associated with germline mutations, which makes pheochromocytoma and paraganglioma solid tumors with a high heritability rate. A genomic characterization study by The Cancer Genome Atlas (TCGA) group, analyzing a cohort of 173 patients, showed that PPGLs can be driven by either germline, somatic or fusion gene mutations in 27%, 39% and 7% of the cases, respectively [6,7,8]. It has been proposed that all patients with PPGL should be considered for genetic testing, as the incidence of hereditary syndromes in apparently sporadic cases is as high as 35% [9,10]. Currently, more than 20 susceptibility genes have been identified, including at least 12 distinct genetic syndromes, 15 driver genes and an expanding fraction of potential disease modifying genes [11,12]. Thus, the underlying mutations appear to determine the clinical manifestations, such as tumor location, biochemical profile, malignant potential, imaging signature and overall prognosis, that should help to tailor treatment and guidance for follow-up. Moreover, detection of a mutation in an index case and their family members should also help clinicians to implement a pertinent surveillance program to promptly identify tumors and treat patients accordingly [13,14]. Despite our understanding of PPGL genetics and molecular biology, the treatment options, especially against advanced and metastatic PPGLs, remain limited and require a personalized approach. Surgical resection remains the mainstay of treatment. In cases where surgery is not feasible or if tumor dissemination limits the probability of curative treatment, the options for treatment are localized radiotherapy, radiofrequency or cryoablation and systemic therapy, which includes chemotherapy or targeted molecular therapies.

There has been increasing interest in radionuclide therapy, which includes ^131^I-MIBG therapy and recently PRRT (peptide receptor radionuclide therapy) ^177^Lu-DOTATATE [15,16,17]. In terms of chemotherapy, CVD (cyclophosphamide, vincristine and dacarbazine) is one of the most traditional chemotherapy regimens and has been used to treat PPGLs over the past 30 years [18]. New treatments are emerging for patients with advanced/metastatic PPGL. Understanding the molecular signaling and metabolomics of PPGL has led to the development of therapeutic regimens for cluster-specific targeted molecular therapies. Based on TCGA classification for cluster I, antiangiogenic therapy, HIF inhibitors, PARP (polyADP-ribose polymerase) inhibition and immunotherapy are used. For cluster II, mTOR (mammalian target of rapamycin) inhibitors are used. Currently there are no cluster III Wnt signaling targeted therapies for PPGL patients [19].

At present, clinical genetic testing for patients with a suspected hereditary form of PPGL is carried out using a germline genetic panel rather than using one gene at a time. Based upon its lower financial cost, immunohistochemistry (IHC) can be considered for screening purposes, particularly in patients with suspected succinate dehydrogenase complex (*SDHx*) mutations. However, IHC should be interpreted with caution as there is likelihood of false-positive and false-negative results [20].

In this review, we summarize recent advances in the discovery of new genes during the past five years. Additionally, we summarize the latest guidelines by Amar et al. for the diagnosis and surveillance of asymptomatic *SDHx* mutation carriers [21].

## 2. Overview of Genetics on What Is Already Known

The identification of the Krebs cycle in the etiology of PPGLs is a milestone in the field of the genetics of PPGLs. The SDH complex plays a pivotal role in energy metabolism in the Krebs cycle, as well as in complex II of the electron transport chain. Mutations in any of the genes encoding the catalytic enzymes of the pathway can lead to an accumulation of their substrates, resulting in hypoxia-inducible factor (HIF) stability and tumorigenesis [22]. These genes include *SDHA*, *SDHB*, *SDHC*, *SDHD*, *SDHAF2* [23,24], fumarate hydratase (*FH*) [25,26], malate dehydrogenase 2 (*MDH2*) [27,28], hypoxia-inducible factor alpha (*HIF2a*) [29,30,31], prolyl hydroxylase (*PHD*) [32] and some newly discovered genes that will be discussed further in the review (Figure 1). Mutation of the genes involved in the kinase receptor signaling pathway that are known to cause PPGLs are *RET* (REarranged during Transfection), neurofibromin 1 (*NF1*), Myelocytomatosis-Associated factor X (*MAX*), transmembrane protein 127 (*TMEM127*), and Harvey rat sarcoma viral gene homologue (*HRAS*). Genes such as *ATRX* (Alpha Thalassemia/mental Retardation-X linked) that are involved in chromosomal integrity, are also implicated as drivers in the etiology of PPGLs and are associated with aggressive behavior [33]. To better understand the genetics based on signaling pathways, The Cancer Genome Atlas (TCGA) has classified PPGLs into three clinically useful molecular clusters: (1) Pseudohypoxic PPGLs, (2) Kinase signaling PPGLs and (3) Wnt signaling PPGLs [34] (Figure 1).

## 3. Genes Discovered in the Last Five Years

With the expanding genetic landscape of PPGLs, several new genes have been identified recently (Table 1) which can potentially predispose patients to the development of tumors with characteristic biological behaviors.

### 3.1. CSDE1 (Cold Shock Domain Containing E1)

*CSDE1* is a tumor suppressor gene located on chromosome 1p13.2 that encodes CSD1 factor, which is involved in messenger RNA (mRNA) stability, internal initiation of translation, apoptosis and neuronal differentiation [7]. Mutation in this gene results in downregulation of the apoptosis protease activator protein 1 (APAF1), which is a critical factor in cellular apoptosis. In the cohort study of 176 patients with PPGL by Feishbein et al. [6], four tumors containing *CDSE1* mutations were detected. These mutations were somatic: two frameshift and two splice-site mutations that clustered proximally within the gene. Patients carrying this gene presented with sporadic and aggressive disease with recurrence and metastasis [6].

### 3.2. H3F3A (Histone Family Member 3A)

The *H3F3A* gene is located on chromosome 1 and encodes the histone H3.3 protein. Histones are scaffolding proteins and the building blocks of the nucleosome. Mutation of *H3F3A* affects DNA methylation, chromatin epigenetics and remodeling, and nucleosome positioning. The first case of an association of the H3F3A mutation and PPGL and GCT (giant cell tumor) of the bone was reported in a 2013 case report by Iwata et al. [36]. In 2016, Toledo et al. characterized a new cancer syndrome involving PPGL and GCT of the bone caused by post-zygotic mutation of the *H3F3A* gene. They analyzed 43 samples from 41 patients by whole exome or transcriptome sequencing and found a post-zygotic *H3F3A* mutation (c103 G > T, p.Gly34Trp) in three tumors from one patient. That patient had recurrent GCT and bilateral PHEO with no family history and developed bladder and periaortic PGL later. This *H3F3A* mutation was identical to one reported as an oncogenic driver of sporadic GCT (c103 G > T, p.G34W) [35]. With this finding, Toledo et al. obtained and analyzed samples from a patient who had aggressive retroperitoneal PGL with liver metastasis and recurrent GCTs, and identified the same H3F3A mutation.

Other chromatin remodeling genes identified in this study were *SETD2* (sporadic PPGL), *EZH2* (sporadic), *KMT2B*, *KMT2D* (sporadic, germline), *ATRX*, *JMJD1C* and *KDM2B* [23].

### 3.3. UBTF-MAML3 (Upstream Binding Transcription Factor Mastermind-like Transcriptional Coactivator 3)

The Wnt pathway is involved in various developmental processes including cell proliferation, adhesion, motility and differentiation. In 2017, Feishbein et al. first reported the association of *MAML3* fusion genes and *CSDE1* (cold shock domain containing E1) of the Wnt and Hedgehog signaling pathways, with the development of PPGLs [6]. In a cohort of 176 patients with PPGL, 10 were positive for the *UBTF-MAML3* fusion gene. The *UBTF* gene is located on chromosome 17q21.31 and encodes the UBTF protein involved in the expression of ribosomal RNA (rRNA) subunits. Patients carrying this fusion gene show extensive alterations in DNA methylation profiles, predominantly hypomethylation that correlates with mRNA overexpression of target genes. In MAML3 fusion-positive tumors, the Wnt pathway members B-catenin, DVL3 (disheveled segment polarity protein-3) and GSK3 (glycogen synthase kinase-3) are overexpressed; whereas miR-375, which is a negative regulator, is underexpressed [39]. Patients with fusion genes have an increased risk of aggressive and metastatic PPGL [6]. It has been shown that *UBTF-MAML3* fusions are expressed in 7% of human PPGLs and overexpression of MAML3 increases tumorigenicity and invasion. Thus, MAML3 expression can serve as a prognostic marker for aggressive disease [45].

### 3.4. IRP1 (Iron Regulator Protein 1)

IRP 1 is a regulator of cellular iron metabolism. In iron deficient cells, IRP1 depresses HIF2α mRNA translation, leading to its accumulation and increased EPO expression [46]. In 2018, Pang et al. discovered the association of *IRP1* with PPGL in a patient with concomitant polycythemia and PHEO [42]. An investigational 54-gene panel carried out on this patient’s peripheral blood DNA was negative for a genetic mutation. Subsequently, tumor DNA sequencing revealed a somatic loss of function mutation in *IRP1* located on the exon 3 splicing site [42,47].

### 3.5. SLC25A11 (Solute Carrier Family 25 Member 11)

*SLC25A11* is a tumor suppressor gene, whose association with PPGL was first reported in 2018 by Buffet et al., which accounts for approximately 1% of all PPGL cases [40]. *SLC25A11* encodes a carrier protein, malate-oxalate carrier (OGC), mediating malate transport from the cytosol to the mitochondrial matrix in exchange for α-ketoglutarate (αKG), while regenerating NADH in the mitochondrial matrix by the electron transport chain complex I [40,41]. Studies have shown that high levels of aspartate and glutamate due to an *SLC25A11* mutation are potent inhibitors of HIF prolyl hydroxylases, which promote tumorigenesis [48]. Buffet et al. demonstrated that germline mutations in the *SLC25A11* gene are strongly associated with the development of metastatic PPGL as 5% of all metastatic PPGLs in their cohort of 121 patients had underlying germline *SLC25A11* mutations. A malignant phenotype was observed in 5 out of the 7 (71%) cases [40]. Germline *SLC25A11* mutations have been found in 5 out of 30 (17%) patients with single, apparently sporadic, metastatic abdominal PGL. Based upon these data, *SLC25A11* mutations should be considered among the genetic risk factors for metastatic PPGL [49].

### 3.6. DLST (Dihydrolipoamide S-Succinyltransferase)

Remacha et al. first described the connection between PPGL and *DLST* in 2019 [43]. *DLST* encodes the E2 subunit of the mitochondrial αKG complex, catalyzing the conversion of α-KG to succinyl-CoA and C0_2_. Mutation in the *DLST* gene results in depletion of the E2 subunit of the αKGD complex, resulting in impaired enzyme activity. Due to this, αKG accumulates leading to high α-KG/fumarate ratio and dysfunction of the Krebs cycle, thus promoting oncogenesis [43]. Five germline variants have been identified that affected the *DLST* gene in eight unrelated individuals; all except one was diagnosed with multiple PPGLs. The above data, therefore, suggest *DLST* as a susceptibility gene for PPGL [43].

Based on a study by Toledo et al., mutations in chromatin remodeling genes and kinase receptor genes (*MERTK*, *MET*, *FGFR1* as described below) are implicated in PPGL pathogenesis [23].

### 3.7. MERTK (Tyrosine Kinase Protooncogene)

Receptor tyrosine kinase cellular signaling pathways regulate a broad variety of cellular processes driving cell growth, proliferation, differentiation, survival, gene transcription and metabolic regulation. Mutations of genes encoding tyrosine kinase receptors are often associated with cancer development [50]. Toledo et al. detected a germline mutation within the tyrosine kinase domain of the *MERTK* gene (c.2273 G > A, p.Arg758His) in a patient with metastatic PPGL and medullary thyroid carcinoma (MTC) [23]. The association of PPGL and MTC is manifested in multiple endocrine neoplasia type 2 (MEN2), which are usually characterized by mutation in the *RET* gene. However, in this patient, *RET* mutation was not detected. To expand on this finding, sequencing of the *MERTK* kinase domain was carried out in a separate cohort of 136 PPGLs. A germline mutation targeting the same residue, R758C, was identified in a patient with sporadic pheochromocytoma. Moreover, somatic *MERTK* mutations were further reported in two cases of PPGL from TCGA dataset. Thus, these findings are suggestive of the role of *MERTK* mutations in PPGL pathogenesis [11,37,38].

### 3.8. MET (Mesenchymal to Epithelial Transition)

*MET* mutations have been reported in multiple cancers [51]. Toledo et al. reported a case of germline mutation of the MET kinase receptor (c.2416 G > A; p.Val806Met) in a patient with a three-generation family history of PPGL [23]. After sequencing 118 unrelated PPGLs, they identified 15 different samples carrying *MET* variants, both germline and somatic, which supports that *MET* mutations are associated with PPGLs [23].

### 3.9. FGFR1 (Fibroblast Growth Factor Receptor 1)

A somatic mutation in *FGFR1* (c.1638C > A; p.Asn546Lys) was detected in one patient with sporadic PPGL, in a cohort of 130 samples by Toledo et al. [23]. This variant was also detected in samples of PPGL from TCGA dataset [37,38], which suggests the association of *FGFR1* mutations with PPGL.

### 3.10. SUCLG2 (Succinyl Co-A Ligase G2)

An association of *SUCGL2* gene mutation with PPGL development was first reported by Vanova et al. in 2021 [44]. Succinyl-CoA ligase (SUCL) is an enzyme of the TCA cycle responsible for the conversion of succinyl-CoA and ADP/GDP to succinate and ATP/GTP [52]. Succinate, which is a product of this enzyme, is an oncometabolite that is linked to the pathogenesis of PPGL [53,54]. SUCL has two subunits. The α-subunit is encoded by *SUCLG1* and the β-subunit by *SUCLA2* (ATP-forming) or *SUCLG2* (GTP-forming) [55]. Vanova et al. [44] tested 352 patients with apparently sporadic PPGL, using a 54-gene panel developed at the National Institutes of Health that included *SUCLG2*, and found that 15 patients had eight germline variants located within the GTP-binding domain of SUCGL-2. To confirm the causality of this defect, a progenitor cell line, hPheo1, derived from a human PPGL, was used. *SUCLG2* germline variants showed increased succinate levels and reduced SDH activity leading to TCA cycle disruption. The pattern of malignancy rate and biochemical phenotype was similar to *SDHx*-mutated PPGLs [56]. Moreover, the *SUCLG2* manipulated hPheo1 cell confirmed the link between the *SUCLG2* mutation and *SDHx* complex function. Hence, the association of a *SUCLG2* gene mutation with the development of PPGL is proposed based on this study. Large scale studies are needed to discover more cases of *SUCLG2* mutations that can provide detailed information about prevalence, penetrance, biochemical phenotype and relationship with *SDHx* in disease etiology.

## 4. New Screening Guidelines for Asymptomatic *SDHx* Carriers

Germline mutations in *SDHx* comprises approximately 20% of cases of PPGL [10,57,58]. When a *SDHx* pathogenic mutation is identified, genetic counselling is proposed for patients’ first-degree relatives. However, there are no established guidelines on how to screen and then follow up asymptomatic mutation carriers. A recent consensus algorithm was established for initial screening and follow-up of *SDHx* mutation carriers by an international panel of 29 experts from 12 countries in 2020, using the Delphi method [21] (Table 2).

The penetrance of *SDHx*-related PPGL is not firmly established. Studies have shown that *SDHB* has a 8–37% penetrance and *SDHD* has a 38–64% penetrance [21,59]. Out of all *SDHx* mutation carriers, patients with an *SDHB* mutation carry the highest risk for metastatic disease [58,60,61]. Patients with *SDHD* mutations have the highest penetrance. Data regarding *SDHC* and *SDHA* mutations are limited but show lower penetrance than *SDHD* mutation carriers [58,62,63]. After analysis of each gene, experts felt that the data are not strong enough to personalize recommendation for each gene separately but concluded that the recommendation for initial screening and follow-up for all *SDHx* genes is the same, with the exception of age of initiation of tumor screening in childhood. Moreover, since *SDHD* has two modes of inheritance—paternal (*SDHD-pi*), which contributes to the majority of transmissions, and maternal (*SDHD-mi*), which is <5% [64], the grade A recommendation is to perform screening for the presence of a tumor once the mutation (*SDHA*, *SDHB*, *SDHC* or *SDHD-pi*) is identified in an asymptomatic carrier, and to perform genetic screening only when tumor screening is considered.

The recommendation for the age at which screening should be performed is extrapolated from the age of incidence of a tumor in a particular gene. A small number of *SDHB*-related PPGLs have been reported in 6-year-old children [13,60,62,65], which is associated with a high risk of metastatic diseases compared to other *SDHx* mutation carriers [13,66]. Therefore, it is recommended to start the screening of *SDHB* mutation carriers between the age of 6–10 years. For asymptomatic *SDHA*, *SDHC*, *SDHD*-*pi* mutation carriers, the recommendation is to initiate screening from 10–15 years. Tumor screening of all asymptomatic *SDHx* (*A*, *B*, *C*, *D*-*pi*) mutation carriers should be started with a history that must include a questionnaire for signs and symptom assessment, followed by clinical examination including blood pressure measurement.

Biochemical testing for tumor screening should include the measurement of either plasma or urine metanephrines and normetanephrines as this seems to be the best diagnostic test with high sensitivity [67,68]. During childhood, the decision of which test to use can be left up to the feasibility and laboratory expertise. On the other hand, for adults, measurement of plasma free metanephrine and normetanephrine should be preferred over urinary tests. It is not recommended to test additionally for catecholamines and vanillylmandelic acid as they are less reliable than metanephrines and normetanephrines [67,68].

Because a significant number of PPGLs are non-functional, especially *SDHx*-related PPGLs, imaging studies must be performed in all asymptomatic *SDHx* mutation carriers. During childhood, a whole-body MRI (head, neck, thoracic, abdomen and pelvic regions) is used as the first-line imaging for initial tumor screening, including patients requiring sedation for MRI. Ultrasound can be used only for children who cannot tolerate MRI [69]. In adults, a combination of head, neck, abdomen and pelvic MRI and PET-CT is recommended for initial tumor screening. Thoracic MRI is not a requirement for initial screening if the PET-CT is normal but it is recommended for subsequent follow-ups. Functional imaging should not be used as first-line screening in children due to radiation exposure. Dedicated cross-sectional imaging should only be carried out if the PET-CT is abnormal. Moreover, use of ^123^I-MIBG and ^111^In-pentetreotide is also not recommended for initial tumor screening.

As patients carrying an *SDHx* mutation have an increased risk for tumor development throughout their life, regular interval follow-ups are recommended even if initial screening results are negative [70]. The expert consensus recommends an annual physical in all age groups, including screening for signs and symptoms of PPGL with questionnaires. Biochemical testing (serum or urinary metanephrines and normetanephrines) must be carried out every 2 years in childhood and every year in adulthood. In terms of imaging, MRI should be performed every 2–3 years in all age groups and functional imaging is not recommended for follow-up screening. If *SDHx* (*A*, *B*, *C*, *D*-*pi*) mutation carriers remain asymptomatic without evidence of a tumor on interval screening, the screening tests should be performed every 5 years after the age of 70, until 80 years of age.

Special consideration should be given to patients who are planning a pregnancy because of complication risks including pre-eclampsia, gestational diabetes and arrhythmias that increase the mortality risk for both the mother and fetus [71]. Therefore, it is recommended to perform complete screening before planning a pregnancy. *SDHx* mutations have been associated with the development of other tumors including renal cell cancer (RCC), gastrointestinal tumors (GIST) and pituitary adenoma [72,73,74,75]. However, no additional screening imaging is recommended, although these tumors should be searched for during the screening imaging.

## 5. Emerging Molecular Genetics and Future Perspectives

The clinical treatment options for patients with PPGL are increasingly based on the underlying molecular biology, genetic and epigenetic analyses of the tumors. In the past two decades, our understanding in the field of genetics, translational research, metabolomics, peptide receptor-based imaging and treatment, as well as immunotherapy, has greatly increased. However, further investigations are needed to deliver precision-based treatment.

Over the last five years, various human- and rodent-derived cell lines and xenografts have been developed. Yet, they do not fully provide subtype classification of tumors and remain challenging for clinical studies. Frankhauser et al. used “immortalized mouse chromaffin cells” (imCCs), MPC/MTT (mouse pheochromocytoma cells/mouse tumor tissue) spheroids, murine pheochromocytoma cell lines and human pheochromocytoma primary cultures, and identified that the PI3Ka inhibitor BYL719 and the MTORC1 inhibitor everolimus are highly effective at tumor shrinkage at clinically relevant doses [76]. To date, there has only been one human cell line progenitor developed successfully: Pheo1 [77]. Moreover, the classic approaches to cell line development, such as SV40-mediated immortalization and newer approaches such as patient-derived tumor xenografts and tumor organoids, have become important preclinical models. Induced pluripotent stem cells (iPSCs) are worth exploring further in this field [78,79]. A recent discovery of the RS0 cell line in 2020 by Powers et al. is a stepping-stone in the field of cell line development and, by far, seems to be the closest model to *SDHB*-mutated human pheochromocytomas [80]. An intrinsic limitation of this model is that it was developed by using irradiation and it is not excluded that the loss of *SDHB* is due to the bystanders effect. Therefore, further characterization by complementation with WT Sdhb must be carried out in the future.

As multiple genetic abnormalities can be associated with a diagnosis of inherited PPGL, next generation sequencing (NGS) is well-suited for carrying out genetic screening. In order to better understand their differences, the classification published by Toledo et al. in 2017 is worth noting as follows: (i) basic panel (including genes mutated at the germline level with the highest evidence for their involvement in the pathogenesis of PPGL), (ii) extended panel (including basic panel genes along with other candidate susceptibility genes that are mutated at the germline level and are found at a low frequency); (iii) comprehensive panel (including extended panel genes along with genes exclusively mutated at the somatic level and those recently found to be mutated at the germline and/or somatic levels, for which the evidence is still limited) [81].

The basic panel encompasses genes involved in germline mutation such as *VHL*, *SDHx*, *FH*, *MAX*, *NF1*, *RET* and *TMEM127* [82]. The extended panel has functionally relevant genes: *EGLN1*, *EPAS1*, *SDHAF2*, *K1F1B* and *MET*. The comprehensive panel includes other recently identified genes [81]. This development has made genetic screening available and affordable in an individual laboratory. However, this comes with the caveat that the analyses become technically challenging with the risk of errors when attempting to add new genes to existing panels [81,83]. The whole exome sequencing (WES) technique is a method of sequencing only the coding regions of DNA and it has led to the identification of several PPGL susceptibility genes such has *MAX*, *FH*, *MDH2*, *HRAS*, *ATRX* and *KMDT*2D [23,81]. Novel NGS techniques, such as RNA sequencing and DNA methylation, can reveal mutational status and can be used as diagnostic or prognostic biomarkers [82].

Several new biomarkers have been discovered that are helpful in differentiating metastatic from non-metastatic tumors and thus prognostication. Major reduction in expression of one putative lncRNA (long non-coding RNA, GenBank: BC063866) has been re-reported in metastatic *SDHx*-related tumors, which itself is an independent risk factor associated with poor clinical outcomes [84,85]. Other metastatic biomarkers identified are hypermethylation of RDBP (negative elongation factor complex member E) promoter and a six-miRNA signature that co-relate time to disease progression [86,87,88]. Cell-free DNA based methods are becoming more popular for cancer detection, however, no such studies have been performed in PPGL patients.

In terms of immunohistochemical markers, biomarkers such as ATRX, chromogranin B and somatostatin receptor 2A have been reported but need more studies to further characterize their roles for prognostication [33,89,90].

The expanding development in the field of the genetics of PPGL has been translated into clinical practice by the provision of widespread testing for inherited PPGL. Utilization of the knowledge of discovery at a molecular level to enable more personalized strategies for investigation, surveillance and management of affected individuals and their families has not only led to accurate diagnosis and risk prediction but also several challenges. These include improving variant interpretation and reducing the number of variants of uncertain significance (VUS), the need for the development of optimal genotypic-phenotypic protocols that enable both early diagnosis whilst keeping healthcare costs in mind, and lastly, producing targeted therapies for metastatic PPGLs. Comprehensive understanding of molecular biology, genetics and oncogenic pathways will lead to the development of novel targets and therapies, which can potentially help improve the prognosis and survival in patients with PPGL [91].

For metastatic PPGL the treatment options have remained limited. Currently, the practiced standard of therapy includes chemotherapy (CVD scheme or temozolomide monotherapy), radionuclide therapy (^131^I-MIBG, ^177^Lu-DOTATATE), tyrosine kinase inhibitors (sunitinib, cabozantinib) and immunotherapy [92,93,94]. A personalized approach is becoming increasingly popular, in light of a comprehensive understanding of molecular biology. These approaches include ^177^Lu DOTATATE therapy for patients with the expression of SSTR2 (somatostatin receptor 2), particularly in *SDHx*-mutated PPGL (positive [67]Ga-DOTATATE scan); ^131^I-MIBG therapy for patients who have expression of the norepinephrine transporter system and are less likely positive for *SDHx*-mutated PPGL (positive ^123^I-MIBG scan); HIF2-α inhibitors for cluster I PPGLs; PARP inhibitors together with temozolomide (especially for *SDHx*-mutated tumors); PDL1 inhibitors (pembrolizumab); and tyrosine kinase inhibitors for cluster II PPGLs [19]. A very recent study by Tabebi et al. showed that downregulation of *SDHB* gene expression in PPGL resulted in increased GLUD1 (glutamate dehydrogenase) expression and can potentially serve as a biomarker and therapeutic target in *SDHB*-mutated PPGLs [95].

Lastly, machine learning algorithms have begun to be used to predict the mutational status in PPGL [96]. Therefore, the combination of artificial intelligence, genetic and immunohistochemical biomarkers, along with metabolomics and clinical features, will be a useful tool for assessing metastatic risk with high accuracy, suggesting long-term prognosis.

## 6. Conclusions

PPGLs are rare NE tumors with unique molecular landscapes. Cataloging and understanding the germline and somatic mutations associated with PPGLs is a promising approach to understand the clinical behavior and prognosis. Moreover, it can provide guidance on diagnostic strategies and personalized treatments for PPGLs.

## Figures and Tables

**Figure 1 cancers-14-00594-f001:**
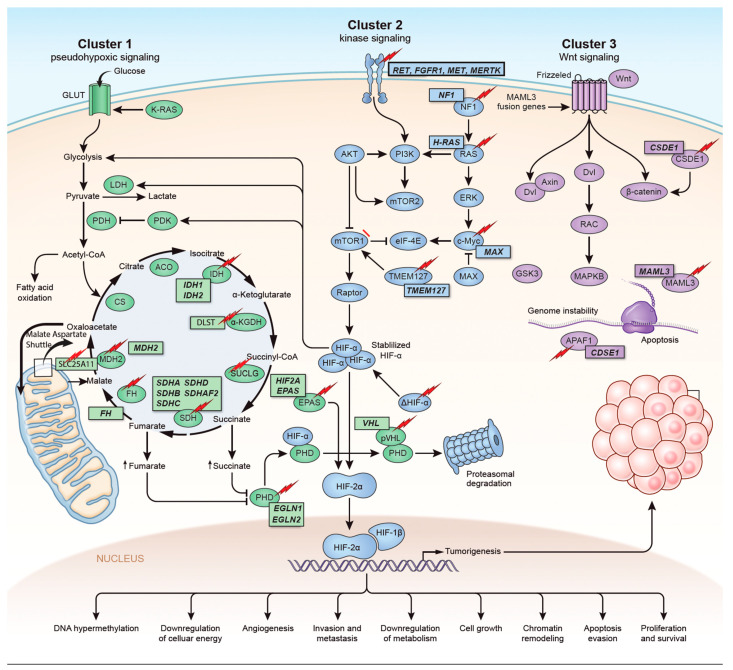
Genetics and molecular pathways for pheochromocytoma and paraganglioma. The genes are classified into three clusters. Cluster I involves mutations in the pseudohypoxic pathway (*SDHx*, *FH*, *MDH2*, *HIF2*, *PHD*, *VHL* and *EPAS*). Cluster II involves mutations in the kinase signaling group (*RET*, *NF1*, *TMEM127*, *MAX* and *HRAS*). Lastly, cluster III includes mutations in the Wnt signaling group (*CSDE1* and *UBTF* fusion at *MAML3*). The new genes discovered (SUCLG2, SLC25A11, DLST, MAPK, MET, MERTK, FGFR1) have been depicted as well. ↑ depicts accumulation of substrate. Adapted from ref. [19].

**Table 1 cancers-14-00594-t001:** Newly discovered in the pathogenesis of PPGLs.

Gene	Year of Discovery	Pathophysiology	Gene Type	Metabolomics	References
*CSDE1*	2016	Tumor suppressor gene involved in mRNA stability and cellular apoptosis	Somatic	Adrenergic	[6,7]
*H3F3A*	2016	Encodes histone H3.3 protein that regulates chromatin formation	Somatic	NA	[35,36]
*MET*	2016	MAPK signaling pathway	Germline, somatic	NA	[23]
*MERTK*	2016	Tyrosine kinase receptor	Germline	NA	[11,37,38]
*UBTF-MAML3*	2017	Unique methylation profile mRNA overexpression involved in Wnt receptor and hedgehog signaling pathways	Fusion	Adrenergic	[6,39]
*SLC25A11*	2018	Encodes malate-oxalate carrier protein of malate-aspartate shuttle	Germline	Noradrenergic	[40,41]
*IRP1*	2018	Cellular iron metabolism regulation	Somatic	noradrenergic	[42]
*DLST*	2019	Encodes E2 subunit of mitochondrial α -KG complex which converts α-KG to succinyl-CoA	Germline	Noradrenergic	[23,43]
*SUCLG2*	2021	Catalyzes conversion of succinyl-coA and ADP/GTP to succinate and ATP/GTP	Germline	Noradrenergic	[44]

**Table 2 cancers-14-00594-t002:** Screening and follow-up guidelines for asymptomatic patients carrying *SDHx* (*A*, *B*, *C*, *D*-*pi*) mutations.

Timeline	Children (<18 Years)	Adults (>18 Years)
Initial screening (*SDHA*, *C*, *D*-*pi*: age 10–15 years, *SDHB*: age 6–10 years)	H/P-questionnaire and BP measurement	H/P-questionnaire and BP measurement
	Biochemical measurements (urinary or plasma M, NM)	Biochemical measurements (plasma M, NM > urinary M, NM)
	Head and neck MRI Thoracic, abdominal and pelvic MRI	Head and neck MRI Abdominal and pelvic MRI
	-	Whole body PET-CT
Follow-up every year	Symptom questionnaire and BP measurement	Symptom questionnaire, BP measurement and biochemical measurements
Follow-up every two years	Biochemical measurements	-
Follow-up every 2–3 years	Head and neck MRI Thoracic, abdominal, and pelvic MRI	Head and neck MRI Thoracic, abdominal and pelvic MRI
Age 80 years	End of follow-up	

H/P—history and physical, BP—blood pressure, M—metanephrine, NM—normetanephrine. Guidelines published by Amar et al. [21].

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
