# Peer review of "New Insights on the Genetics of Pheochromocytoma and Paraganglioma and Its Clinical Implications"

_cancers, 2022, doi:10.3390/cancers14030594_

Round 1

Reviewer 1 Report

The authors provide a very detailed review on the currently known mutational spectrum of PPGL and the resulting consequences for the screening strategies as well as future perspectives.

This is a very systematic, well written, and comprehensive review addressing the in general highly interesting and still emerging field of tumor heritability and predisposition.

Overall, the knowledge gain through this review in addition to previous publication, in particular the cited review by Ilanchezhian et al. 2020 (see also adapted Figure 1) might be limited.

In addition, few aspects need to be addressed, before the manuscript can be considered for publication in cancers.

Major comments:

  1. As already addressed, the overall knowledge gain provided by this review and thus its impact and novelty might be limited, especially considering earlier publications, for example the cited review by Ilanchezhian et al. 2020 (see also adapted Figure 1). It could be helpful to address this aspect directly and emphasize the additional information provided by this review in comparison to earlier reports.

  1. In the introduction, a paragraph summarizing treatment options of these diseases is missing. This should be added.

  1. Also in the introduction, the authors summarize that 40% of cases are associated with germline mutations and a further 30-40% show somatic diver mutations (lines 46-49). Are there any ongoing investigations, for example whole exome sequencing approaches to detect disease-causing mutations in the remaining 20% of cases?

  1. The different gene mutations are very well described. However, many of these alterations are very rare and it appears difficult to imagine that general recommendations can be derived from these cases. The manuscript would probably benefit if the authors could critically discuss this aspect.

  1. In the section addressing the emerging molecular genetics and future perspectives, the authors state, that “future investigations are needed to deliver precision-based treatment” (lines 302-303). However, they refer almost exclusively to surveillance and screening recommendations and don`t address therapeutic approaches. Here, the manuscript could benefit from a paragraph on clinical and therapeutic consequences emerging from the increasing genetic knowledge.

Minor comments:

  1. In the section addressing the new screening guidelines, the authors discuss that “ultrasound can be used for only for children who cannot tolerate MRI” (lines 267-268). Does this statement refer to children who need sedation/anesthesia to perform an MRI?
  2. In the emerging molecular genetics and future perspectives section, the paragraph referring to the use of next generation sequencing (lines 321-324) sounds very general and basic and could probably be omitted. In addition, it seems that the authors erroneously used the term “single-cell”, here.

Author Response

Major comments:

  1. As already addressed, the overall knowledge gain provided by this review and thus its impact and novelty might be limited, especially considering earlier publications, for example the cited review by Ilanchezhian et al. 2020 (see also adapted Figure 1). It could be helpful to address this aspect directly and emphasize the additional information provided by this review in comparison to earlier reports.

We thank the Reviewer for this interesting comment. In this review particularly, we have tried to highlight only the most important aspects and updates in the field of PPGL over the past five years, which we consider as providing readers with new/up-to-date information. We have expanded upon the genes discovered in the past five years including the novel SUCLG2 gene mutation which was reported very recently in 2021. Moreover, based on discovery of new genes involved in pathogenesis of PPGL, we have updated that information in the figure taken from Ilanchezhian et al.[19] as well. We have also discussed the recently published guidelines for screening and evaluation for asymptomatic SDHx carriers in the review and have critically discussed the future perspectives in this field.

  1. In the introduction, a paragraph summarizing treatment options of these diseases is missing. This should be added.

We appreciate this excellent suggestion by the Reviewer. We have incorporated the following paragraph in the introduction section of the revised manuscript.

"Despite our understanding of PPGL genetics and molecular biology, the treatments options especially against advanced and metastatic PPGL remain limited and require personalized approach. Surgical resection remains the mainstay of the treatment. In cases where surgery is not feasible or if tumor dissemination limits probability of curative treatment, the options for treatment are localized (radiotherapy, radiofrequency or cryoablation) and systemic therapy which includes chemotherapy or targeted molecular therapies. There has been increasing interest in radionuclide therapy which includes 131I-MIBG therapy and recently PRRT (peptide receptor radionuclide therapy) 177Lu-DOTATATE 15-17. In terms of chemotherapy, CVD (cyclophosphamide, vincristine and dacarbazine) is one of the most traditional chemotherapy regimens and has been used to treat PPGL over the past 30 years 18. New treatments are emerging for patients with advanced/metastatic PPGL. Understanding molecular signaling and metabolomics of PPGL has led to development of therapeutic regimens for cluster specific targeted molecular therapies. Based on TCGA classification, for cluster I, antiangiogenic therapy, HIF inhibitors, PARP (polyADP-ribose polymerase) inhibition and immunotherapy are used. For cluster II, mTOR (mammalian target of rapamycin) inhibitors are used. Currently there are no Wnt signaling targeted therapies for PPGL patients19.

  1. Also in the introduction, the authors summarize that 40% of cases are associated with germline mutations and a further 30-40% show somatic diver mutations (lines 46-49). Are there any ongoing investigations, for example whole exome sequencing approaches to detect disease-causing mutations in the remaining 20% of cases?

We thank the Reviewer for this excellent remark. To our knowledge, there has not been any report that directly reported the ongoing investigations of the remaining 20% cases by whole exome sequencing in the literature. However, recently, we have attempted to perform either exome sequencing or evaluation of gene in custom-made gene panels (e.g. the NIH gene panel) in over 350 patients with apparently sporadic PPGLs that did not show any germline or somatic mutation using well-known gene panels provided by commercial companies. The recent discovery of SUCLG2 gene is one example. Currently, we have another discovery related to the fusion gene (the manuscript has been submitted) and several additional new potential candidate genes that could be involved in the pathogenesis of these tumors. We are also working with other medical centers or institutions to perform genetic analysis of apparently sporadic PPGLs. 

  1. The different gene mutations are very well described. However, many of these alterations are very rare and it appears difficult to imagine that general recommendations can be derived from these cases. The manuscript would probably benefit if the authors could critically discuss this aspect.

The Reviewer is very correct that some gene mutations are very rare. Therefore, in the present manuscript, although mentioned, we do not go into many details how they could be used in clinical-making decision algorithm but rather we describe them. We have added the statement, that in case that there is no mutation found in common well-known PPGL susceptibility genes, and there is still a high suspicion that other rare PPGL susceptibility genes could play a role, to discuss such a situation with experts in the field whether further genetic testing is warranted. We consider this approach as very fair.

  1. In the section addressing the emerging molecular genetics and future perspectives, the authors state, that “future investigations are needed to deliver precision-based treatment” (lines 302-303). However, they refer almost exclusively to surveillance and screening recommendations and don`t address therapeutic approaches. Here, the manuscript could benefit from a paragraph on clinical and therapeutic consequences emerging from the increasing genetic knowledge.

We appreciate this suggestion from the Reviewer. We have added the following paragraph discussing clinical and therapeutic consequences from genetic knowledge to the revised manuscript:

The expanding development in the field of genetics of PPGL has been translated into clinical practice by the provision of widespread testing for inherited PPGL. Utilization of knowledge of discovery at molecular level to enable more personalized strategies for investigation, surveillance and management of affected individuals and their families has not only lead to accurate diagnosis and risk prediction but also several challenges to improve variant interpretation and reduce the number of variants of uncertain significance (VUS), need for development of optimal genotypic-phenotypic protocols that enable both early diagnosis keeping healthcare costs in mind and lastly targeted therapies for metastatic PPGL. Comprehensive understanding of molecular biology, genetics and oncogenic pathways will lead to development of novel targets and therapies which can potentially help improve prognosis and survival in patients with PPGL92.

For metastatic PPGLs, the treatment options have remained limited. Currently, the practiced standard of therapy includes chemotherapy (CVD scheme or temozolomide monotherapy), radionuclide therapy (131I-MIBG, 177Lu-DOTATATE), tyrosine kinase inhibitors (sunitinib, cabozantinib) and immunotherapy93-95. Personalized approach is becoming increasingly popular in the light of comprehensive understanding of molecular biology. These approaches include 177Lu DOTATATE therapy for patients with expression of SSTR2 (somatostatin receptor 2), particularly in SDHx-mutated PPGLs (positive 68Ga-DOTATATE scan), 131I-MIBG therapy for patients who have expression of norepinephrine transporter system, less likely positive for SDHx mutated PPGL (positive 123I-MIBG scan), HIF2-a inhibitors for cluster 1 PPGL, PARP inhibitors together with temozolomide (especially for SDHx mutated tumors), PDL1 inhibitors (pembrolizumab), tyrosine kinase inhibitors for cluster 219. A very recent study by Tabebi et al. showed that downregulation of SDHB gene expression in PPGL resulted in increased GLUD1 (glutamate dehydrogenase) expression and can potentially serve as a biomarker and therapeutic target in SDHB mutated PPGL96.

Minor comments:

  1. In the section addressing the new screening guidelines, the authors discuss that “ultrasound can be used for only for children who cannot tolerate MRI” (lines 267-268). Does this statement refer to children who need sedation/anesthesia to perform an MRI?

We thank the Reviewer for this excellent comment. We agree with the Reviewer that there may be some practical implications and challenges to obtain MRI for children and they may need sedation/anesthesia. In those cases when sedation is needed, MRI is preferrable over ultrasound which is inferior to any CT or MRI in general. This information has been incorporated into the revised manuscript. We again thank the Reviewer for pointing out this important clinical scenario.

  1. In the emerging molecular genetics and future perspectives section, the paragraph referring to the use of next generation sequencing (lines 321-324) sounds very general and basic and could probably be omitted. In addition, it seems that the authors erroneously used the term “single-cell”, here.

We appreciate the Reviewer's comment, and we apologize for the error. We have omitted the sentence "Next generation sequencing (NGS) where many genes can be analyzed simultaneously has replaced conventional single-cell sequencing over the past decade. Moreover, in targeted NSG only specific coding region of gene is targeted and used to perform genetic screening" and replaced with the following paragraph "As multiple genetic abnormalities can be associated with a diagnosis of inherited PPGLs, next generation sequencing (NGS) is well-suited for carrying out genetic screening. In order to better understand their differences, the classification published by Toledo et al. (2017) is worth noting as follows: i) basic panel (including genes mutated at the germline level with the highest evidence for their involvement in the pathogenesis of PPGLs), ii) extended panel (including basic panel genes along with other candidate susceptibility genes that are mutated at the germ line level and are found at a low frequency); iii) comprehensive panel (including extended panel genes along with genes exclusively mutated at the somatic level and those recently found to be mutated at the germline and/or somatic levels for which the evidence is still limited)".

Reviewer 2 Report

Sakashi Jhawar and coauthors in the manuscript “New Insights on the Genetics of Pheochromocytoma and Paraganglioma”  report an overview on the genetics of PPGLs with the newly discovered genes and discussing the latest guidelines on surveillance of asymptomatic SDHx mutation carriers. The paper  is interesting and well-written.

Minor revisions:

  1. Line 21: in the “Simple Summary” section , “SDHx” must be written in italic.

  1. Line 47: what do the authors mean with the term “PCPG”? Typing error? Please, explain or correct it.

  1. Line 51: the authors state “Currently, more than 20 pathogenic mutations have been identified [...]”, but, to my knowledge, many more than 20 pathogenic mutations have been reported in at least 20 known susceptibility genes. Please, clarify this point.

  1. Line 74: please replace “ADHAF2” with “SDHAF2”.

  1. Line 88: the figure 1 shows those genes related to the three main clusters for PPGLs. “PDH1 and PDH2” are incorrect, as they were called as “PHD1 and PHD2”. However, it is noting that their current approved symbol is EGLN2 and EGLN1, respectively. Please, use the updated HGCN nomenclature both in the figures and in the maintext.

  1. Line 97: Table 1, the authors should separate the new susceptibility genes that are mutated at the germline level from those one mutated at the somatic level, I might add a new column in the table 1.

  1. Line 113: please replace “be” with “by”.

  1. Line 117 and 120: please replace the entire mutation description in the brackets (c103 G>T, p.G34W) with (c.103G>T, p.Gly34Trp).

  1. Line 160: concerning SLC25A11, the authors state: “ Buffet et al. Demonstrated that SLC25A11 gene mutations are strongly associated with the development of metastatic PPGL as 5% of all metastatic PPGLs in their cohort of 121 patients had underlying SLC25A11 mutations.” Which type of mutations do the authors mean? Germline or somatic? Please, add it in the previous sentence.

  1. Line 185: please replace “p.R758H” with “p.Arg758His”.

  1. Line 197: please replace “p.V8.6M” with “p.Val806Met”.

  1. Line 202: please replace “p.N546K” with “p.Asn546Lys”.

13. Line 324: The authors mention that three types of targeted NGS panels are used. In order to better understand their differences, the classification published by Toledo et al. (2017) should be reported, as follows: i) basic panel (including genes mutated at the germline level with the highest evidence for their involvement in the pathogenesis of PPGLs), ii) extended panel (including basic panel genes along with other candidate susceptibility genes that are mutated at the germ line level and are found at a low frequency); iii) comprehensive panel (including extended panel genes along with genes exclusively mutated at the somatic level and those recently found to be mutated at the germline and/or somatic levels for which the evidence is still limited).

  1. Line 333: Please replace the article citations [44, 81] with the original articles discovering the new PPGL susceptibility genes through the application of WES technique.

  1. Line 338: please replace “LncRNA” with “lncRNA”.

Author Response

  1. Line 21: in the “Simple Summary” section, “SDHx” must be written in italic.

We thank the Reviewer for noticing this error. We have corrected in the revised manuscript.

  1. Line 47: what do the authors mean with the term “PCPG”? Typing error? Please, explain or correct it.

We thank the Reviewer for pointing out this mistake. We have corrected "PCPG to PPGL"          (pheochromocytoma and paraganglioma) in the revised manuscript.

  1. Line 51: the authors state “Currently, more than 20 pathogenic mutations have been identified [...]”, but, to my knowledge, many more than 20 pathogenic mutations have been reported in at least 20 known susceptibility genes. Please, clarify this point.

Thank you for this excellent comment. We apologize for this sentence. The correction has been made in the revised manuscript. There are over 20 known susceptibility genes associated with PPGL.

  1. Line 74: please replace “ADHAF2” with “SDHAF2”.

We thank the Reviewer for this comment. We apologize for the typo. The correction has been made in the revised manuscript.

  1. Line 88: the figure 1 shows those genes related to the three main clusters for PPGLs. “PDH1 and PDH2” are incorrect, as they were called as “PHD1 and PHD2”. However, it is noting that their current approved symbol is EGLN2 and EGLN1, respectively. Please, use the updated HGCN nomenclature both in the figures and in the maintext.

We thank the Reviewer for this excellent remark. We have replaced "PDH1 and PDH2" by "EGLN2 and EGLN1", respectively in the figure as well as the main text of the revised manuscript.

  1. Line 97: Table 1, the authors should separate the new susceptibility genes that are mutated at the germline level from those one mutated at the somatic level, I might add a new column in the table 1.

We thank the Reviewer for this excellent suggestion. We have added a separate column in the table of the type of gene- somatic, germline, fusion gene. Please see the following:

Gene

Year of discovery

Pathophysiology

Gene Type

Metabolomics

CSDE1

2016

Tumor suppressor gene involved in mRNA stability and cellular apoptosis

Somatic

Adrenergic

H3F3A

2016

Encodes histone H3.3 protein that regulates chromatin formation

Somatic

NA

MET

2016

MAPK signaling pathway

Germline, somatic

NA

MERTK

2016

Tyrosine kinase receptor

Germline

NA

UBTF-MAML3

2017

Unique methylation profile mRNA overexpression involved in Wnt receptor and hedgehog signaling pathways

Fusion

Adrenergic

SLC25A11

2018

Encodes malate-oxalate carrier protein of malate-aspartate shuttle

Germline

Noradrenergic

IRP1

2018

Cellular iron metabolism regulation

Somatic

noradrenergic

DLST

2019

Encodes E2 subunit of mitochondrial α -KG complex which converts α-KG to succinyl-CoA

Germline

Noradrenergic

SUCLG2

2021

Catalyses conversion of succinyl-coA and ADP/GTP to succinate and ATP/GTP

Germline

Noradrenergic

  1. Line 113: please replace “be” with “by”.

We thank the Reviewer for this comment. We have made the correction in the revised manuscript.

  1. Line 117 and 120: please replace the entire mutation description in the brackets (c103 G>T, p.G34W) with (c.103G>T, p.Gly34Trp).

We thank the Reviewer for this suggestion. The changes have been made in the revised manuscript.

  1. Line 160: concerning SLC25A11, the authors state: “Buffet et al. Demonstrated that SLC25A11 gene mutations are strongly associated with the development of metastatic PPGL as 5% of all metastatic PPGLs in their cohort of 121 patients had underlying SLC25A11 mutations.” Which type of mutations do the authors mean? Germline or somatic? Please, add it in the previous sentence.

We appreciate this interesting comment and clinically useful suggestion from the Reviewer. We have made the following changes in the revised manuscript- "Buffet et al. Demonstrated that germline SLC25A11 gene mutations are strongly associated with the development of metastatic PPGL as 5% of all metastatic PPGLs in their cohort of 121 patients had underlying germline SLC25A11 mutations.

  1. Line 185: please replace “p.R758H” with “p.Arg758His”.

We appreciate the comment from the Reviewer. The changes have been made in the revised manuscript

  1. Line 197: please replace “p.V8.6M” with “p.Val806Met”.

We thank the Reviewer for the suggestion. The changes have been made in the revised manuscript.

  1. Line 202: please replace “p.N546K” with “p.Asn546Lys”.

We have made the correction in the revised manuscript

  1. Line 324: The authors mention that three types of targeted NGS panels are used. In order to better understand their differences, the classification published by Toledo et al. (2017) should be reported, as follows: i) basic panel (including genes mutated at the germline level with the highest evidence for their involvement in the pathogenesis of PPGLs), ii) extended panel (including basic panel genes along with other candidate susceptibility genes that are mutated at the germ line level and are found at a low frequency); iii) comprehensive panel (including extended panel genes along with genes exclusively mutated at the somatic level and those recently found to be mutated at the germline and/or somatic levels for which the evidence is still limited).

We thank the Reviewer for this excellent suggestion and paragraph explaining about the types of NGS. We have incorporated this information in the revised manuscript.

  1. Line 333: Please replace the article citations [44, 81] with the original articles discovering the new PPGL susceptibility genes through the application of WES technique.

We thank the Reviewer for the comment. Following references have been added to the revised manuscript- [23, 82]

  1. Line 338: please replace “LncRNA” with “lncRNA”.

Thank you for the suggestion. We have corrected in the revised manuscript.

Reviewer 3 Report

Jhawar and colleagues provide an overview of recent discoveries of various mutations linked to pheochromocytomas and paragangliomas. Laudably, the review is of high quality and is a very good read. It includes a very well researched bibliography and is up-to-date. The manuscript is very well written and there are only minor mistakes that need to be adjusted (amongst others abbreviations not spelled out prior first use). I have attached a scan of my handwritten annotations - those should be self-explanatory.

I suggest reconsidering the title of the manuscript as -in my opinion- it falls a little short. There is a considerable section that focuses on guidelines in asymptomatic SDHx mutation carries, which is very important but not adequately reflected by the title. Maybe "New Insights on the Genetics of Pheochromocytoma and Paraganglioma and their Clinical Implications" (?)

In summary, this is a very good manuscript that very well deserves to be published after minor corrections.

Author Response

  1. Title- The Reviewer suggested adding "Clinical implications" to the title "New Insights on the Genetics of Pheochromocytoma and Paraganglioma".

We appreciate this interesting comment from the Reviewer. We have changed the title of the revised manuscript to "New Insights on the Genetics of Pheochromocytoma and Paraganglioma and its Clinical Implications"

  1. Line 39- The Reviewer suggested to add per million per year to the sentence "The incidence of PHEOs and PGLs (collectively PPGLs) is estimated at approximately 2-8 cases per million."

We thank the Reviewer for pointing it out. We have replaced the error "2-8 cases per million" with "2-8 cases per million per year".

  1. Line 48- The Reviewer suggested to explain the abbreviation "PCPG"

We thank the Reviewer for pointing out this error. We have replaced "PCPG" with "PPGL" in the revised manuscript.

  1. Line 66- The Reviewer recommended to remove the extra space after Amar et al.

We thank the Reviewer for this comment. We have omitted the extra space after Amar et al. in the revised manuscript.

  1. Line 148- The Reviewer suggested to correct the sentence with "Polycythemia Vera"

We appreciate this comment from the Reviewer. We have changed the sentence to "Polycythemia" and removed "vera" as the mechanism is unknown at the present time.

  1. Line 173- The Reviewer suggested to omit the extra space.

We thank the Reviewer for this comment. We have omitted the extra space in the revised manuscript.

  1. Line 192- The Reviewer suggested to expand the term "TCGA".

We appreciate the Reviewer for this remark. We mentioned the term TCGA previously in line 83 where we also explained it's full term as "the cancer genomic atlas".

  1. Line 225, section 4. New screening guidelines for asymptomatic SDHx carriers, where the Reviewer commented "nice, but doesn’t fit the title/subject of the paper well".

We appreciate the Reviewer for this excellent comment. We agree with the Reviewer that it may not be directly related to genetic pathophysiology of PPGL. We therefore changed the title of the review to "New insights in genetics of pheochromocytoma and paraganglioma and its clinical implications". Moreover, we also added some clinical and therapeutic aspects as well in the maintext.

  1. Line 269- The Reviewer suggested to omit the extra space in the sentence.

We thank the Reviewer for this comment. We have removed the extra space in this sentence in revised manuscript.

  1. Line 294, Table 2. The Reviewer suggested to rearrange the alignment of the table.

We thank the Reviewer for this comment. We have rearranged the table in the revised manuscript.

  1. Line 331- The Reviewer suggested to remove an extra "period".

We appreciate this comment from the Reviewer. We have omitted "period" between progenitor and developed.

  1. Line 321- The Reviewer pointed out the error "NSG" instead of "NGS".

We thank the Reviewer for pointing out this error. We have replaced the paragraph "Next generation sequencing (NGS) where many genes can be analyzed simultaneously has replaced conventional single-cell sequencing over the past decade. Moreover, in targeted NSG only specific coding region of gene is targeted and used to perform genetic screening" and replaced with the following paragraph "As multiple genetic abnormalities can be associated with a diagnosis of inherited PPGLs, next generation sequencing (NSG) is well-suited for carrying out genetic screening. In order to better understand their differences, the classification published by Toledo et al. (2017) is worth noting as follows: i) basic panel (including genes mutated at the germline level with the highest evidence for their involvement in the pathogenesis of PPGLs), ii) extended panel (including basic panel genes along with other candidate susceptibility genes that are mutated at the germ line level and are found at a low frequency); iii) comprehensive panel (including extended panel genes along with genes exclusively mutated at the somatic level and those recently found to be mutated at the germline and/or somatic levels for which the evidence is still limited)".

  1. Line 328- The Reviewer recommended to omit the extra space in this sentence.

We thank the Reviewer for this comment. We have removed the extra space in this sentence.

  1. Line 348- The Reviewer pointed out a grammatical mistake as Machine learning algorithms have beginning to be used and suggested to replace with "begun" instead of "beginning".

 We appreciate the Reviewer's comment. We have replaced "beginning" with "begun" in the revised manuscript.

Round 2

Reviewer 1 Report

Dear Sirs,

From my perspective, the manuscript has been sufficiently improved to warrant publication in Cancers.